# The COVID-19 Pandemic Affected Hepatitis C Virus Circulation and Genotypic Frequencies—Implications for Hepatitis C Prevention, Treatment and Research

**Julio Daimar Oliveira Correa**  **and José Artur Bogo Chies** * 

Post-Graduation Program in Genetics and Molecular Biology, Biosciences Institute, Federal University of Rio Grande do (UFRGS), Av. Bento Gonçalves, 9500, Campus do Vale Prédio 43323, Porto Alegre 91501-970, Brazil; juliodaimar@gmail.com
* Correspondence: jabchies@terra.com.br

**Abstract:** Hepatitis C is regarded as a global health issue caused by hepatitis C virus (HCV) infection. HCV is targeted for elimination by 2030 as a global public health goal. However, the COVID-19 pandemic has changed human circulation and prevented access to diagnostics and treatment to many other diseases, including hepatitis C. COVID-19 impacted HCV global elimination efforts with implications not fully comprehended yet. The high genetic variability in HCV makes the development of vaccines and pan-genotypic drug therapies a difficult task. Changes in the dynamics of HCV impose new challenges for public health and opportunities for future research. Meta-analysis, the follow up of new cases and sampling of HCV patients compared with previously available data are options for investigating the possible changes. The determination of HCV genotypes and subtypes is important for understanding viral dynamics and treatment; therefore, the changes in genotype and subtype prevalences can directly affect such processes. Recent results in the literature already suggest changes in HCV dynamics during the COVID-19 pandemic, both considering viral circulation and differential genotypic frequencies in distinct geographic areas. In this context, we propose a further examination of these trends using different approaches to provide support for the hypothesis that the COVID-19 pandemic affected HCV circulation, since these findings would have important implications for hepatitis C prevention, treatment and research.

**Keywords:** hepatitis C; HCV; circulation; COVID-19; pandemic

## 1. Introduction

Hepatitis C virus (HCV) infection is the cause of hepatis C liver disease and a global health issue. The World Health Organization (WHO) has targeted HCV elimination by 2030 as a global public health goal, introducing global targets for the care and management of HCV in 2016 [1]. However, this goal was settled before the COVID-19 pandemic. Ever since, reports of changing patterns of circulating viral infections, such as in influenza and dengue, during the COVID-19 pandemic have been reported [2,3].

### 1.1. Impact of COVID-19 on Other Viral Infections

The circulation of several respiratory viruses, such as influenza, has been globally disrupted since the emergence of COVID-19 and the introduction of public health and social measures aimed at reducing severe acute respiratory syndrome coronavirus 2 (SARS-CoV-2) transmission [2]. International surveillance reports showed historically low levels of infections during the influenza season in multiple countries [4–7]. There has been a delay or absence in several traditional seasonal influenza milestones in the period of the years 2020, 2021 and 2022 in countries like Canada, Australia, New Zealand and Saudi Arabia [2,4,7]. Changes in influenza activity and circulating subtypes during the COVID-19 outbreak in China have also been reported [8]. This study claims that influenza subtypes changed

from A-dominant to B/Victoria-dominant. Mobility dropped below baseline levels after the introduction of COVID-19 restrictions [2,9,10], which impacted infections relying on human contact. This is probably true not only considering respiratory infections, but also infections for which transmission occurs via exposition to contaminated blood and other body fluids, such as HCV.

During the period where the COVID-19 pandemic was defined as a Public Health Emergency of International Concern (PHEIC) by the WHO, the tests for and detection of influenza and other infectious diseases significantly decreased as the focus was COVID-19 itself [4,9]. In fact, the emergence of COVID-19 resulted in a decrease in patients receiving HCV testing and treatment worldwide [11–13]. It has been stated that COVID-19 impacted HCV global elimination efforts, with prediction models suggesting that the impact extends beyond the direct morbidity and mortality associated with exposure and infection [14].

*1.2. HCV Diversity*

HCV exhibits a high genetic diversity, even when compared to HIV-1 [15]. Importantly, the diversity of HCV is characterized by regional variations in genotype prevalence. This feature poses a challenge for the development of vaccines and pan-genotypic drug therapies [15,16]. HCV is a positive-sense RNA virus containing a single large open reading frame encoding a polyprotein of approximately 3010 amino acids [17]. Phylogenetic and sequence analyses of the whole viral genome are used to classify HCV strains into genotypes and subtypes [15]. The phylogenetic analysis of HCV genomes revealed that sequences fall into different clusters [16], and a standardized nomenclature was proposed in 1994 [16,18]. Presently, HCV is classified into eight genotypes (genotypes 1–8) and each genotype has different subtypes (example: 1a, 1b, 1c), approximately 93 in total [19].

As previously mentioned, the global distribution of HCV genotypes is regionally specific and genotype 1 is predominant in most regions [16,20]. HCV genotype 1 is found in the frequencies of 92.6% in the Caribbean, 90.9% in Andean Latin America, 89.2% in Central Europe, 87% in Southern Latin America, 75.8% in high-income North America, 74.9% in high-income Asia Pacific, 71.7% in Central Latin America, 69.3% in Tropical Latin America, 66.6% in Central Asia, 65.5% in Eastern Europe, 57% in Southeast Asia and 54.2% in Australasia [15,20]. Genotype 2 is found in the frequencies of 24.5% in high-income Asia Pacific, 23% in Wester Sub-Saharan Africa, 19.3% in Central Latin America and 18.2% in Southeast Asia [15,20]. Genotype 3 is found in the frequencies of 71.6% in South Asia and 39.2% in Australasia [15,20]. Genotype 4 is found in the frequencies of 97.6% in Central Sub-Saharan Africa, and 65.3% in North Africa and Middle East [15,20]. Genotype 5 is found in the frequency of 58.8% in Southern Sub-Saharan Africa [15,20]. Genotype 6 is found at its highest frequency of 16.2% in East Asia [15]. Genotypes 7 and 8 are the rarest HCV genotypes [19].

An important fact to highlight regarding HCV genotype prevalence is the regional fluctuations in frequencies of the most common genotypes [20]. These regional fluctuations are illustrated in Figure 1 which shows a graphical representation of the frequencies of HCV genotypes in different regions based on a review by Messina et al. [15]. Table 1 expands the data from Figure 1, showing the overall HCV prevalence in the different regions. Also, an important point to highlight is the fact that data of HCV genotypes prevalence for most countries are prior to 2019 [21]. In fact, data from Figure 1 and Table 1 refer to the frequencies observed in 2015. This point, instead of being viewed as a limitation, can be highlighted as an opportunity for new research with new data, making it possible to compare prevalence before and after the COVID-19 pandemic.

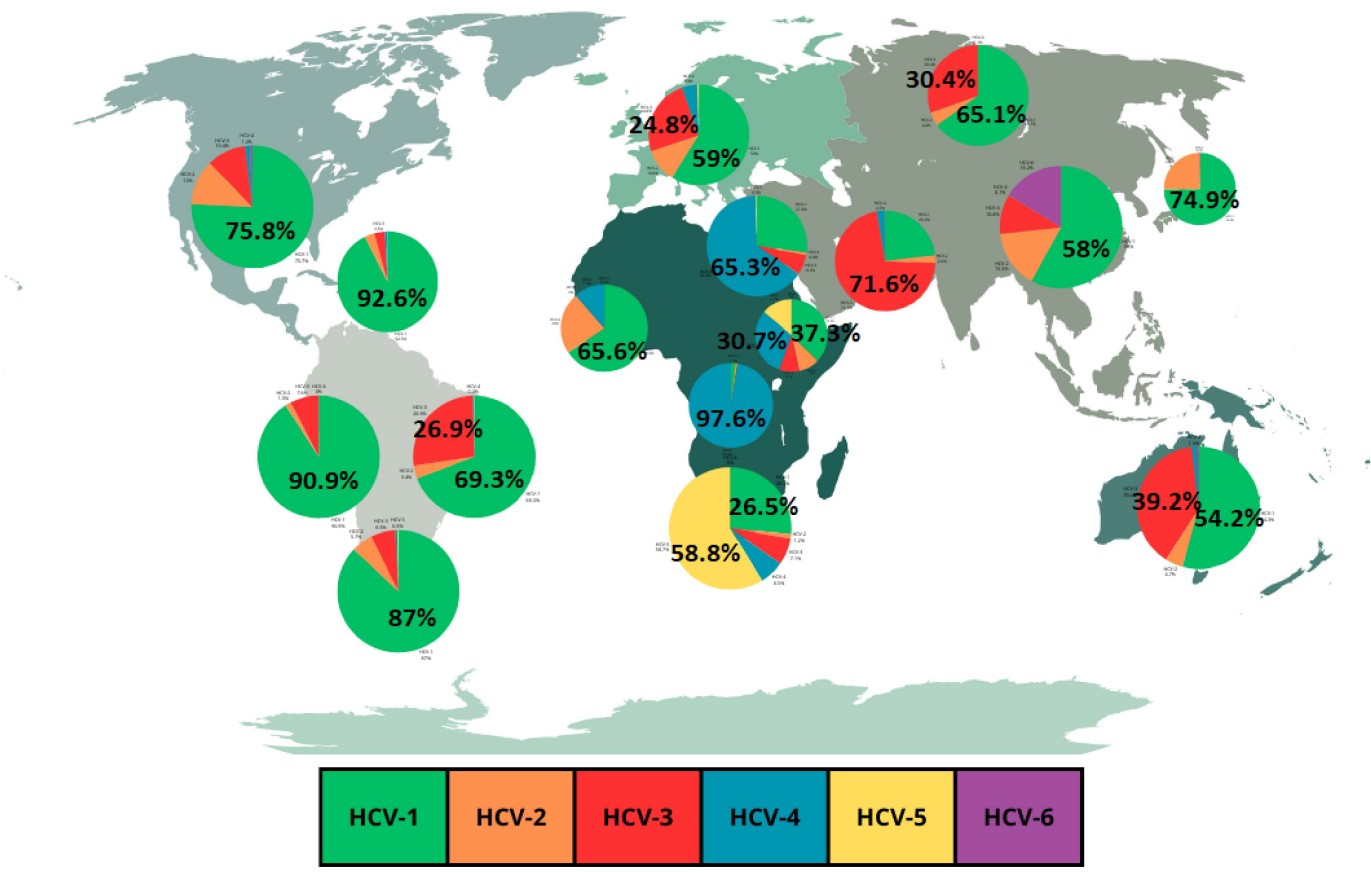

**Figure 1.** Relative prevalence of most common HCV genotypes by regions. Data adapted from Messina et al. [15]. For the complete data see Table 1. The size of the discs has no correlation with the total HCV incidence in regions. Figure 1 template was downloaded from Canva under a Free Media License.

**Table 1.** Frequencies of HCV genotypes by region according to Figure 1.

| Main Region | HCV-1 | HCV-2 | HCV-3 | HCV-4 | HCV-5 | HCV-6 |
|---|---|---|---|---|---|---|
| North America | 75.8% | 12% | 10.4% | 1.2% | 0.1% | 0.6% |
| Caribbean | 92.6% | 3.2% | 3.5% | 0.8% | 0 | 0 |
| Tropical Latin America | 69.3% | 3.4% | 26.9% | 0.3% | 0.1% | 0 |
| Andean Latin America | 90.9% | 1.5% | 7.6% | 0 | 0 | 0 |
| Southern Latin America | 87% | 5.7% | 6.5% | 0.5% | 0.3% | 0 |
| Southern Africa | 26.5% | 1.2% | 7.1% | 6.5% | 58.8% | 0 |
| Central Africa | 1.7% | 0.8% | 0 | 97.6% | 0 | 0 |
| Eastern Africa | 37.3% | 92.% | 9.1% | 30.7% | 13.7% | 0 |
| Western Africa | 65.6% | 23% | 0 | 11.3% | 0.1% | 0 |
| North Africa/Middle East | 27.3% | 0.8% | 6.3% | 65.3% | 0.3% | 0 |
| Western Europe | 59% | 10.8% | 24.8% | 4.9% | 0.5% | 0 |
| Eastern Europe | 65.1% | 4.4% | 30.4% | 0.1% | 0 | 0 |
| South Asia | 23.2% | 2.4% | 71.6% | 2.5% | 0.1% | 0.1% |
| East Asia | 58% | 15.3% | 10.4% | 0.1% | 0 | 16.2% |
| Asia Pacific | 74.9% | 24.5% | 0.6% | 0 | 0 | 0 |
| Australasia | 54.2% | 4.7% | 39.2% | 1.3% | 0 | 0.5% |

## 2. The Hypothesis

We hypothesize that HCV circulation has been affected by the COVID-19 pandemic. In this same direction, alterations in the relative prevalence of the most common HCV genotypes could be observed in distinct regions. Also, other aspects of HCV natural history

could be impacted, and this impact should be verified in the course of the following years. As a virus with a slower cycle of infection, compared to influenza, the impact on HCV circulation needs to be assessed over a longer period.

Reduced mobility can reduce the circulation of HCV. In contrast, reduced testing and treatment can contribute to the maintenance of the virus in the population. The prevalence of circulating HCV genotypes may be similarly affected as observed for influenza. Changes in the dynamics of HCV prevalence and distribution impose new challenges for the global public health goal of HCV elimination by 2030 for hepatitis C treatment and opportunities for future research.

## 3. Evaluation of the Hypothesis

This hypothesis could be evaluated with a series of different approaches, including a meta-analysis of reported cases before, during and after the COVID-19 pandemic outbreak. This approach would provide a more precise estimate of the effect on a larger scale, gathering results of individual cases reported around the world to increase the generalizability of the results [22].

Also, a follow up of new cases of HCV infection, from the diagnostics to treatment, evolution and possible resolution of infection can lead to a comparison between infection prior to the COVID-19 pandemic and new cases of HCV infection. The COVID-19 pandemic can be considered to have exerted a huge systemic impact, as public health systems from all over the world concentered their focus on it [23]. As such, significant differences between the dynamics of infection and progression of HCV before and after this global event can be associated with the impact of the COVID-19 pandemic.

The sampling of HCV patients for new direct studies on infection dynamics, viral and host genetics, as well as response to treatment can be implemented to test this hypothesis. This will help identify and understand the differences after the COVID-19 pandemic. It is important to take into consideration that different human populations have different levels of access to treatment. This is a problem, especially for developing countries, and poses a different kind of challenge for the global public health goal of HCV elimination by 2030. New studies can address this issue as well.

## 4. Conclusions

### 4.1. Limitations of the Hypothesis

Differently from influenza, HCV is not a respiratory infection, being primarily transmitted via contact with contaminated body fluids. Nevertheless, despite different transmission modes, all infections rely on human contact. Another important point is that hepatitis C can also lead to a chronic infection. In spite of such behavior, HCV will be affected in our globalized and dynamic world, with new events eventually altering HCV dynamics. In fact, testing this hypothesis could provide an opportunity for a better understanding of the dynamics of infection diseases in a general way.

Access to effective treatment in the form of direct-acting antiviral (DAA) tablets, especially in developed countries, represents another important situation to take into account, since effective treatment can reduce HCV prevalence independently of other factors. Continued screening allied with treatment has proven to be effective in controlling and reducing HCV prevalence, as recently demonstrated in Egypt [24] and in a study in prisons in Spain [25]. However, the impacts on HCV genetic diversity due to the COVID-19 pandemic could also be a potential source for antiviral resistance, directly disrupting current treatment from a global perspective.

### 4.2. Final Remarks

The assumption that there were changes in HCV infection dynamics, transmission and circulation regarding virus genotypes opens the possibility for impacts on HCV genetic studies and treatment. Taking into consideration that HCV has high genetic diversity, this possibility is always present. Changes in HCV infection dynamics, transmission and

circulation can lead to the selection of existing genotypes, or even the emergence of new subtypes. In fact, new subtypes reported recently, including a novel genotype 4 subtype in Saudi Arabia, shed light on the ongoing evolution and diversity of the virus [19].

As previously mentioned, HCV is classified according to genotypes. In general, the determination of a HCV genotype and subtype is important for understanding viral evolution, transmission and epidemiology, as well as the selection of appropriate treatments [17].

The fact that HCV displays high genetic diversity at the genotype and subtype levels creates the necessity to characterize novel subtypes and understand the potential impact of said subtypes on treatment outcomes [26]. The potential impacts of the COVID-19 pandemic in novel subtypes cannot be overlooked. New studies on the genetics and treatment of HCV highlight these impacts by comparing new results with data available prior to the COVID-19 pandemic. Even minor alterations in genotype distribution/frequencies, as well as the observation of non-previously common genotypes or subtypes in certain regions, can directly affect hepatitis C prevention and treatment, and open opportunities for new research.

As mentioned earlier, the COVID-19 pandemic led to a decrease in HCV testing and treatment, as highlighted in a recent study [27]. This could potentiate an increase in HCV genetic variability due to the maintenance of diverse viruses in infected but untested and untreated individuals. This scenario opens the possibility for the selection of new traits, including drug resistance. Finally, the COVID-19 pandemic also boosted the development of mRNA vaccines [28]. Conventional mRNA vaccines are affordable and can be rapidly manufactured, displaying high flexibility and adaptability to address emerging variants and outbreaks [28]. In fact, a study for an HCV mRNA vaccine candidate has already been reported by a north American group [29]. The potential of mRNA vaccines should be considered another factor that can impact HCV circulation in the future.

In conclusion, the hypothesis that the COVID-19 pandemic impacted HCV circulation should be rigorously tested and evaluated due to the important implications for hepatitis C prevention, treatment and research, which promotes reaching the WHO goal of HCV elimination by 2030.

**Author Contributions:** J.D.O.C.: Conceptualization, Methodology, Validation, Formal analysis, Investigation, Writing—Original Draft, and Visualization. J.A.B.C.: Writing—Review & Editing, Supervision. All authors have read and agreed to the published version of the manuscript.

**Funding:** This research received no external funding.

**Institutional Review Board Statement:** Not applicable.

**Informed Consent Statement:** Not applicable.

**Data Availability Statement:** Data are contained within the article.

**Conflicts of Interest:** The authors declare that they have no competing financial interest or personal relationships that could influence the work reported in this paper.

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
