# Peer review of "The COVID-19 Pandemic Affected Hepatitis C Virus Circulation and Genotypic Frequencies—Implications for Hepatitis C Prevention, Treatment and Research"

_epidemiologia, doi:10.3390/epidemiologia5020011_

Round 1
Reviewer 1 Report
Comments and Suggestions for Authors
I would suggest to the authors to make some comments about the importance of HCV screenings, as well as the significance of using the latest antiviral treatments. There are studies conducted in the Spanish prison population where they have achieved nearly micro-elimination in a highly vulnerable population. Also, countries like Egypt have recently achieved a significant decrease in prevalence rates thanks to the two mentioned pillars: screenings and treatment with the new antivirals.
Reviewer 2 Report
Comments and Suggestions for Authors
Dear authors, the hypothesis about COVID and HCV is well written, please in :Line 77 review by Messina et al. add ‘’.’’
Line 119 add .
Reviewer 3 Report
Comments and Suggestions for Authors
Overall, the article provides a comprehensive overview of the potential impact of the COVID-19 pandemic on the circulation of Hepatitis C virus (HCV) and its implications for prevention, treatment, and research. Here are some comments and suggestions for improvement:
Title: The title is informative, clearly indicating the focus of the article. However, it could be more concise for clarity. Perhaps consider rephrasing it to something like "The Impact of the COVID-19 Pandemic on Hepatitis C Virus Circulation: Implications for Prevention, Treatment, and Research."
Abstract: The abstract succinctly summarizes the main points of the article. However, it could be strengthened by including specific findings or implications discussed in the subsequent sections of the article.
Introduction: The introduction provides a good overview of the global significance of HCV infection and the WHO's goal for its elimination by 2030. However, it could benefit from a clearer transition to the discussion on the impact of the COVID-19 pandemic.
Content Organization: The content is logically organized into sections. However, consider providing subsection headings within each section to improve readability and navigation.
Data Presentation: When presenting data on the prevalence of HCV genotypes in different regions, ensure that the information is clearly presented and easy to interpret. Consider using tables or figures to visually represent the data and highlight regional variations.
Clarity and Precision: Some sentences in the article could be clarified for better understanding. For example, in the section discussing the hypothesis, consider revising the sentence "Given what we know about how other viral infections were impacted by COVID-19 pandemic and prediction models, it is highly possible that HCV circulation have also been affected by COVID-19 pandemic" for clarity and precision.
Conclusions: The conclusions effectively summarize the key findings and implications discussed in the article. However, consider adding a brief discussion on the potential limitations of the study and avenues for future research.
Reviewer 4 Report
Comments and Suggestions for Authors
The manuscript "How has COVID-19 pandemic impacted HCV circulation? Implications for Hepatitis C prevention, treatment and research" shows an analysis of an interesting current topic. However, I have some comments.
1. Please revise English.
2. I suggest the authors to change some keywords in order to have better search results.
3. There are already a few reports of metadata analysis showing a decrease of HCV treatment and testing after COVID pandemic, hence it would be important if authors could mention these studies in order to enrich the discussion of the text.
4. Please improve quality of Figure 1. Also, I suggest to include the overall HCV prevalence in the different regions in the image.
5. It would be very interesting if authors could discuss whether the alterations in HCV testing and treatment due to COVID may result in increased genetic variability and therefore in possible antiviral resistance.
6. Additionally, since there was a rapid development of mRNA vaccines technologies for COVID, it would be interesting to add a paragraph in the manuscript discussing about the possibility of implementation of this technology for HCV vaccine design.
Comments on the Quality of English Language
Please revise English throughout the text. Some phrases are too long and need grammar checking.
Reviewer 5 Report
Comments and Suggestions for Authors
The view entitled "What effect has the COVID-19 pandemic had on the spread of HCV? Julio Daimar Oliveira Correa and José Artur Bogo Chies' "Implications for Hepatitis C Prevention, Treatment, and Research" does a good job of explaining how COVID 19 has affected HCV circulation globally, and the evidence they offer supports their conclusions. As a result, the current version may be published in the journal.
Round 2
Reviewer 4 Report
Comments and Suggestions for Authors
All issues have been addressed
